# The Regional Differences in Game-Play Styles Considering Playing Position in the FIBA Female Continental Basketball Competitions

**DOI:** 10.3390/ijerph17165827

**Published:** 2020-08-12

**Authors:** Zongpeng Zhai, Yongbo Guo, Yuanchang Li, Shaoliang Zhang, Hongyou Liu

**Affiliations:** 1School of Physical Education and Sports Science, South China Normal University, Guangzhou 510006, China; zhaizongpeng@m.scnu.edu.cn (Z.Z.); guoyongbo@m.scnu.edu.cn (Y.G.); liyuanchang@m.scnu.edu.cn (Y.L.); 2Division of Sports Science and Physical Education, Tsinghua University, Beijing 100084, China

**Keywords:** playing positions, female basketball, FIBA continental championships, generalized linear mixed model

## Abstract

The aim of this study was to identify regional differences based on playing position in terms of the technical performances among FIBA Female Continental Basketball Championships by controlling the influence of situational variables including the game outcome, game type, teams and opponent quality. The samples comprised of 9208 performance records from 471 games in the America, Africa, Asia and Europe Championships during 2013–2017 and were collected and analyzed by generalized mixed linear modeling. Our study highlighted that, although positional differences were clear among different continental championships, it is worth noting that African guards, forwards, and centers made more turnovers (TOV) compared with the corresponding positional players from other continental championships. In addition, European guards presented the lowest number of steals (STL) compared with African (ES = 0.28), Asian (ES = 0.21), and American guards (ES = 0.24). The results provide coaches to have a better understanding of game-play styles among FIBA Female Continental Basketball Competitions, which could optimize the development of female basketball and the selection and recruitment of female players at the international level.

## 1. Introduction

Studies in basketball have documented the anthropometric, technical and physical demands of athletes at various playing positions [1,2,3,4,5]. This information is useful for player assessment, recruitment, selection, and prescription of training to ultimately guide long-term development of players [6,7]. Although some studies explored the differences between male and female basketball in terms of technical and physical performances in different levels of basketball competitions, the majority of studies are still related to the development and evaluation of men’s basketball which has led to the performance profiles of female basketball lagging far behind men’s basketball. Currently, an updated and influenced study about the regional differences in technical performances among FIBA Female Continental Basketball Championships was conducted by Madarame [8], but positional differences in this study are not yet confirmed.

In fact, playing positions have been well established based on specific function and characteristics on the court. For example, centers typically take advantage of rebounds and guards show a better performance on assists and three-point shots [3,9]. Mangine et al. [10] divided basketball players into two positions (backcourt and frontcourt) to identify the relationship between visual tracking speed and reaction time on special basketball technique. Moreover, Pion et al. [11] applied the linear and nonlinear technique to analyze and demonstrate that body mass and sprint ability were the main factors to discriminate the five positions on the court. Notably, it is the most common way to classify basketball players into three positions as guards, forwards and centers in the FIBA international basketball competitions [2,12,13,14,15,16,17,18].

The available studies have identified the differences of female basketball players considering playing positions from anthropometric, physical, physiological and biomechanical perspectives. Scanlan et al. [19] made a comparison with technical and physical demands among playing positions in Australian female basketball players. The results indicated that backcourt players performed more dribbling, less standing, less walking and less running than frontcourt players. Likewise, Delextrat and Cohen [2] found that guards have better performance than centers in Wingate Anaerobic test, torque of knee extensors, single-leg jumps, suicide runs and agility T-tests, while forwards achieved better performance than centers in torque of the knee extensors. Similarly, Štrumbelj et al. [5] made an evaluation about physiological parameters (oxygen consumption, carbon dioxide production, pulmonary ventilation breath by breath, respiratory quotient and oxygen pulse) based on a graded shuttle run test where no significant differences were found between playing positions in elite female basketball players. From a biomechanical view, each player has their own shooting style because the differences of length proportions between upper body segments [20,21]. Pehar et al. [22] identified the differences of jump capacity between guards, forwards and centers and the results showed that non-significant differences between playing positions in standing broad jump (SBJ) and countermovement jump (CMJ) but guards and forwards had a better performance in four running jump achievement, reactive strength index (RSI) and repeated reactive strength ability (RRSA) due to the lower body mass. Additionally, Sindik and Jukić [4] found that guards had higher efficiency in assists and three-point shots than centers whereas centers performed better in two-point shots and offensive and defensive rebounds. Also, Sampaio et al. [3] stated that the differences in technical performances that are attributed to the anthropometric characteristics among different basketball leagues. However, to our knowledge, there were no studies to explore the differences in technical performances of female basketball considering playing position among America, Africa, Asia and Europe Championships.

According to the above consideration, the aim of the present study was to identify regional differences in technical performances considering playing position among FIBA Female Continental Basketball Championships by controlling the influence of situational variables including the game outcome, game type, teams and opponent quality. We hypothesized that female guards, forwards and centers from different continental championships may present different technical advantage. The results of this study can be applied into the selection and recruitment of suitable players for ultimate success in the international female basketball competitions.

## 2. Materials and Methods

### 2.1. Sample

All the publicly available data were obtained from the official FIBA Female Continental Basketball Championships (https://archive.fiba.com/). The sample comprised of 9208 performance records from 471 female basketball games among America, Africa, Asia and Europe in 2013, 2015 and 2017. The game-related statistics were transformed to per-minute statistics (original statistics/min × 40 min) according to players’ playing time on the court. Additionally, the players who played within five minutes or only one game were also removed from the sample [3]. The variables selected (Table 1) in the present study were consistent with those employed by Zhang et al. [23]. The study was conducted according to the ethical guidelines of the authors’ affiliated institutions but did not require Ethics Committee approval because a non-interventional design was used whereby all analyzed data were de-identified and available in the public domain.

### 2.2. Validity and Reliability

Two experienced basketball performance analysts (both had an experience of more than 5 years acting as a basketball coach/performance analyst) observed and coded 20 randomly chosen games from the aforementioned sample. The coded results were compared to the collected data from the website. For the variables of assists, the obtained Intra-Class Correlation Coefficients (ICC) was 0.83, while for the rest of the variables, the ICC was 1.0, which showed very acceptable reliability.

### 2.3. Statistical Analysis

Generalized mixed linear modelling was realized in the software of SAS Studio 3.6 (SAS Institute Inc, Cary, NC, USA) using the code of Proc Glimmix. Fixed effects in the modelling included the variables of playing position, match outcome, match type, team strength, and opponent strength. Player identity and team identity were included as random effects to deal properly with the repeated measurements. For each of the dependent variables (technical variables), an independent Poisson regression was run and results were output by the group of continental championships.

Continents, match result, and match type were added as foul-level (Africa, Asia, America and Europe), two-level (won and lost) and two-level (balanced and unbalanced: point difference under and above 10 points) nominal predictors into the modelling. The effect of team strength and opponent strength was estimated by creating a new variable name “rank difference”, which took the log of the ranking of the team divided by the ranking of the opponent team in the Championships [24].

The non-clinical magnitude-based inference was employed to assess the uncertainty in the true effects undertaking a published spreadsheet [25]. Estimated effects and their confidence intervals were standardized by dividing the between-player SD derived from the modelling. The following scale was used to evaluate the magnitude of the effects: trivial < 0.2 < small < 0.6 < moderate < 1.2 < large < 2.0 < very large [26]. If the 90% confidence limits included simultaneously substantial positive and negative values (> 0.2 and < 0.2), the effect was deemed unclear. Clear substantial or trivial effects (whichever likelihood was greater) were reported using the following quantitative scale: most unlikely < 0.5% < very unlikely < 5% < unlikely < 25% < possibly < 75% < likely < 95% < very likely < 99.5% < most likely [26].

## 3. Results

Descriptive statistics of technical variables of female basketball players of different playing positions in the four continental championships (America, Africa, Asia and Europe) are presented in Table 2. The differences in the mean counts of each technical variable considering playing positions among female continental championships are presented in Figure 1, Figure 2 and Figure 3.

### 3.1. Difference of the Technical Performances between Centers (Figure 1)

African centers performed less 2 ptM and 2 ptA than European (ES = 0.28; ES = 0.23) and Asian centers (ES = 0.31; ES = 0.42), but more TOV than European (ES = 0.25) and American centers (ES = 0.23). Moreover, American centers secured more DREB than European (ES = 0.29) and Asian centers (ES = 0.39).

### 3.2. Difference of the Technical Performances between Forwards (Figure 2)

Asian (ES = 0.25; ES = 0.37) and African (ES = 0.28; ES = 0.34) performed more 2 ptM and 2 ptA than European forwards. In addition, African forwards recorded more FTA than European (ES = 0.48), Asian (ES = 0.31), and American forwards (ES = 0.32) while they made more TOV than European (ES = 0.42), Asian (ES = 0.23), and American forwards (ES = 0.29). However, they made less 3 ptM and 3 ptA than European (ES = 0.30; ES = 0.25), Asian (ES = 0.37; ES = 0.45), and American forwards (ES = 0.21; ES = 0.22).

### 3.3. Difference of the Technical Performances between Guards (Figure 3)

European guards performed less 3 ptA than Asian guards (ES = 0.30) whereas they committed more PF than Asian guards (ES = 0.30). Furthermore, African and American guards made more TOV than Asian (ES = 0.43; ES = 0.32) and European guards (ES = 0.46; ES = 0.35). American guards made more DREB and AST than African guards (ES = 0.25; ES = 0.27). European guards presented the lowest value of STL compared with African (ES = 0.28), Asian (ES = 0.21), and American guards (ES = 0.24).

The current study identified regional differences in technical performances considering playing positions among FIBA Female Continental Basketball Championships by controlling the influence of situational variables including the game outcome, game type, teams and opponent quality. In agreement with our hypothesis, our study noted that African guards, forwards, and centers made more TOV compared with the corresponding positional players from other continental championships. In addition, European guards presented the lowest number of STL compared with African, Asian, and American guards. Therefore, this study can assist coaching staff and performance analysts to have a deep understanding of positional differences in the FIBA international female continental basketball competitions. Furthermore, it provided valued references for coaches to select and recruit suitable players preparing for national teams.

## 4. Discussions

### 4.1. The Differences of Technical Performance between Centers

Centers play an crucial role in executing two-point shots because they usually stay near the basket with the natural advantage of physique and specific tactic leading to a higher field-goal shooting percentage [3,27,28]. However, our results showed that African centers performed less 2 ptM and 2 ptA than European and Asian centers, meaning that African centers may participate less in inside plays or shooting with low efficiency in the paint and middle range. Coaches are supposed to pay more attention to develop African centers’ post-up skills and improve shooting efficiency. Personnel scouts should note this characteristic of African female centers in order to make an informed decision in the draft process. Furthermore, our result showed that American centers secured more DREB than European centers, which further identified the prior studies from Erčulj and Štrumbelj [29], Paulauskas et al. [30] and Zhang et al. [31] pointed out that American players had better stature and fitness with high efficiency in terms of defensive rebounds. In fact, defensive rebounds have become one of the most key performance indicators (KPIs) for ultimate success in almost all levels of basketball competitions [1,28,30,32]. Zhang suggested that this KPI may reflect high level performances associated with (i) game pace, with more defensive rebounds indicating more fast-break ball possessions; (ii) players physical characteristics with stronger and taller players securing more rebounds, likely due to superior stretch shortening cycle, jumping performances; and (iii) effective communication with defensive strategy [1]. Therefore, selection of American female centers, and incorporation of a faster playing style, may provide greater match success at the international elite female level.

### 4.2. The Differences of Technical Performance between Forwards

Traditionally, forwards usually undertake multiple tasks on the court (not only inside play but also outside play). Currently, with the popularity of “small ball” in the NBA (e.g., Houston Rockets and Golden State Warriors) [31,33], coaches or managers prefer players who do not truly have a favored role as a result of their malleability, able to play in two, three or in some cases even more areas of the court and still maintain a consistently high standard in each [3,31,33,34]. However, our study found that African forwards tend to make more than 2 ptM and 2 ptA compared with European forwards and they highlighted the lowest 3 ptM and 3 ptA among the four regions. These results may reveal a typical strategy that African forwards were played in a traditional manner and lack of high-versatility traits. This game-style of African forwards could make a negative impact on the whole team offensive strategy because it is hard for them to space the floor and occupy their defenders. In addition, our study was consistent with previously reported findings that African forwards tend to utilize free throws to scoring due to more energetic and active behavior in isolation (lay-ups and paint penetrations) which could lead to more physical contacts to free throws [30]. From the way of scoring, coaches should more develop the versatility of African forwards in order to take the key function in the given situation during the game play.

### 4.3. The Differences of Technical Performance between Guards

Our results showed that European guards performed less 3 ptA than Asian guards whereas they committed more PF than Asian guards, which were linked with the previous studies suggested that European guards tend to be more cautious before shooting three points, especially in a lower ball possession because the competition level in Europe maintains a higher game intensity [7,8]. In addition, Sampaio and Janeira [27] suggested that fouls were an effective way to ball possession recovery. Similarly, Ibáñez et al. [7] also emphasized that European coaches tend to use fouls tactics to interfere with the leading team’s game rhythm and fixed technical tactics during European basketball games. It could give a tip to those players who play against with European teams such as accelerating the game’s pace and avoiding being fouled. Furthermore, as a vital factor for game results, AST were often executed by guards and regarded as one of the main indicators to measure the level of rapport among players as well as the integrity of a team’s offense [1,7,30,35]. To accomplish AST, players should not only be proficient with the skills of ball-handling and passing but also the ability of observation and prediction, choosing the best timing and the most appropriate approach to pass to teammates [1,36,37]. In addition, our study found that guards play a key role in high-intensity competitions (e.g., Europe and America) and perform higher number of AST than that of Africa, which was partly supported by Madarame [8], Paulauskas et al. [30] and García et al. [38] who suggested that offense in Europe was well-organized due to the higher number of AST in European competitions. Therefore, Limiting the European and American guards in the game might be the prior defensive strategy for their opponents.

KPIs need to be especially emphasized in our study, TOV and STL. Our study found that African players from all of positions made more TOV compared with the corresponding positional players from other continental championships which was linked to the studies of Sampaio et al. [3] and Madarame [8] suggested that the game pace and the unorganized manner of play in Africa were the main reasons that lead to the emergence of turnovers. In addition, it is worth noting that European guards had the lowest number of steals and the differences were clear in the pairwise comparison. Consistent with previous results (European forwards had the least steals), it confirmed again that the conservative defensive style was executed by European teams as a broad consensus [7]. On the other hand, it indicated that European guards performed well in ball handling, penetration, moving, observation, communication, passing and avoiding turnovers, which may have exposed the deficiency of African guards, forwards and centers in above aspects because they made more TOV compared with the corresponding positional players from other continental championships. As a consequence, opponents could put more pressure to force African players to turnovers. For African players themselves, they should avoid unnecessary dribble, decrease the rhythm of game and be patient in offense. There were some limitations of this study that should be acknowledged. First, the results were based on regular performance indicators which means that further studies are encouraged to excavate some data of tactical behavior such as isolations, screens, hand-offs, cuts, and transitions. Second, all the comparisons in this study were made specifically between the groups by a non-direct competition. Future work may wish to compare these variables in a simultaneous situation (e.g., Olympic game or Basketball World Cup). Third, future research is also encouraged to consider the physical or physiology indicators like somatotype, stature and body composition as the variables which have been proven to have a significant impact on player’s performance. Finally, the current research only considers the difference of players with corresponding positions, which means that future studies can develop a comparison within and between different positions across different regional competitions.

## 5. Conclusions

In summary, although most of the differences were small or moderate, some game-play styles could still be identified. African players had relatively low performance when considering the indicators on passing, shooting and ball-handling. Specifically, African players showed a weakness in offense considering the number of 2-point field goals and turnovers. African forwards performed a traditional style with a high number of 2-point shots but a low efficiency. American players lay more emphasis on defensive rebounds. European players showed a broad conservative strategy of defense considering a low number of steals and turnovers. The results identified the differences of positional technique in female competitions, which may provide innovative perspectives on the development of modern female basketball games, as well as designing more specific training plans for coaches and players.

## Figures and Tables

**Figure 1 ijerph-17-05827-f001:**
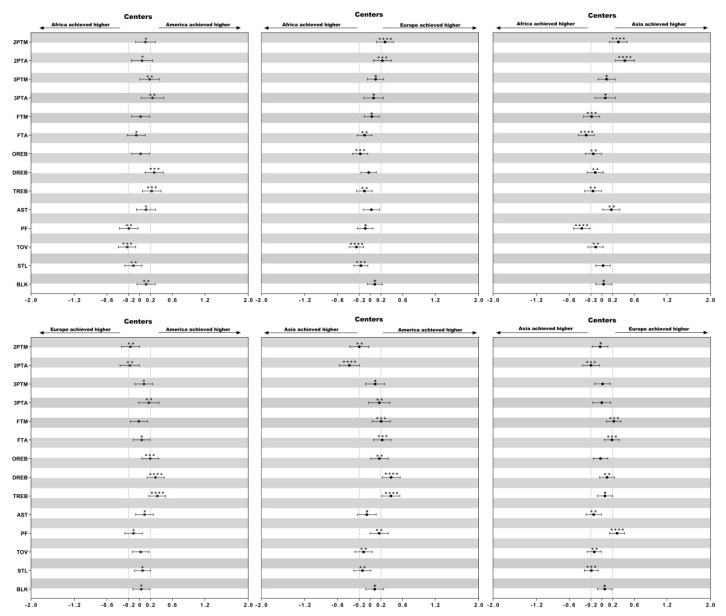
Standardized differences of technical performances in centers among different continental championships. Bars present 90% confidence intervals. Asterisks indicate the likelihood for the magnitude of the true difference as follows: * possible; ** likely; *** very likely; **** most likely. Asterisks located in the area between −0.2 and 0.2 denote for trivial differences.

**Figure 2 ijerph-17-05827-f002:**
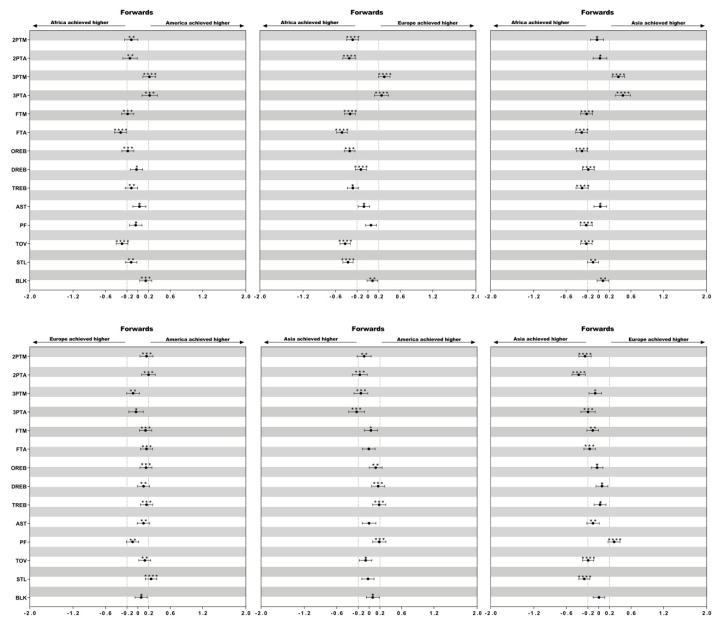
Standardized differences of technical performances in forwards among different continental championships. Bars present 90% confidence intervals. Asterisks indicate the likelihood for the magnitude of the true difference as follows: * possible; ** likely; *** very likely; **** most likely. Asterisks located in the area between −0.2 and 0.2 denote for trivial differences.

**Figure 3 ijerph-17-05827-f003:**
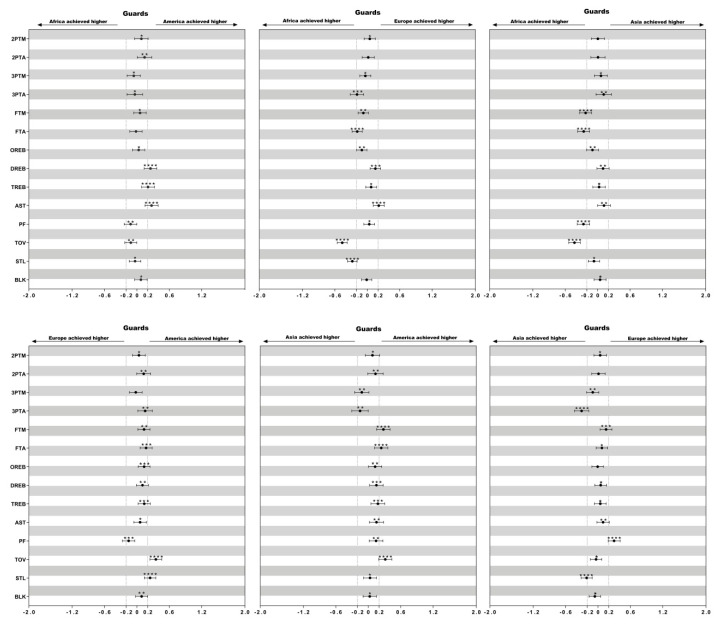
Standardized differences of technical performances in guards among different continental championships. Bars present 90% confidence intervals. Asterisks indicate the likelihood for the magnitude of the true difference as follows: * possible; ** likely; *** very likely; **** most likely. Asterisks located in the area between −0.2 and 0.2 denote for trivial differences.

**Table 1 ijerph-17-05827-t001:** The variables selected in current study.

Variables	Definition
Two-point Made (2 ptM)	The number of two-point field goals that a player has successfully made
Two-point Attempted (2 ptA)	The number of two-point field goals that a player has attempted
Three-point Made (3 ptM)	The number of three-point field goals that a player has successfully made
Three-point Attempted (3 ptA)	The number of three-point field goals that a player has attempted
Free throws Made (FTM)	The number of free throws that a player has successfully made
Free throws Attempted (FTA)	The number of free throws that a player has attempted
Offensive Rebounds (OREB)	The number of rebounds that a player has secured, while they were on offence
Defensive Rebounds (DREB)	The number of rebounds that a player has secured, while they were on defense
Total Rebounds (TREB)	The total number of rebounds that a player has collected
Assists (AST)	An assist occurs when a player completes a pass to a teammate that directly leads to a successful field goal
Personal Fouls (PF)	The total number of fouls that a player has committed
Turnovers (TOV)	A turnover occurs when the teams on offence loses the ball to the defense
Steals (STL)	A steal occurs when a defensive player takes the ball from a player on offence, causing a turnover from offensive players
Blocks (BLK)	A block occurs when an offensive player attempts a shot, and a defensive player tips the ball, blocking their chance to score

**Table 2 ijerph-17-05827-t002:** Descriptive statistics of technical performances considering playing positions among four continental championships.

Variable	Africa			Asia			America			Europe		
	Guards	Forwards	Centers	Guards	Forwards	Centers	Guards	Forwards	Centers	Guards	Forwards	Centers
2 ptM	2.8 ± 2.86	4.46 ± 3.57	4.5 ± 3.49	2.79 ± 2.85	4.19 ± 3.58	5.52 ± 4.21	3.05 ± 2.72	3.84 ± 3.36	4.87 ± 3.82	2.98 ± 2.49	3.41 ± 2.93	5.49 ± 3.52
2 ptA	7.33 ± 4.86	10.21 ± 5.8	10.65 ± 5.58	7.13 ± 4.94	9.9 ± 5.85	13 ± 7.5	7.94 ± 4.67	9.02 ± 5.5	10.92 ± 5.82	7.34 ± 4.19	8.1 ± 4.63	11.92 ± 5.41
3 ptM	1.45 ± 2.12	0.7 ± 1.36	0.03 ± 0.3	1.5 ± 2.08	1.28 ± 2.03	0.19 ± 0.76	1.31 ± 1.8	1.03 ± 1.81	0.3 ± 0.96	1.31 ± 1.68	1.22 ± 1.73	0.19 ± 0.74
3 ptA	5.13 ± 4.58	2.74 ± 3.3	0.39 ± 0.92	5.15 ± 4.48	4.38 ± 4.56	0.69 ± 1.6	4.71 ± 3.78	3.5 ± 4.13	1.14 ± 2.2	4.13 ± 3.12	3.78 ± 3.45	0.65 ± 1.5
FTM	2.26 ± 3.12	3.11 ± 3.57	2.85 ± 3.53	1.53 ± 2.49	2.33 ± 3.12	2.29 ± 2.96	2.44 ± 3.03	2.42 ± 3.22	2.93 ± 3.2	2.13 ± 2.89	2.08 ± 2.74	3.12 ± 3.44
FTA	3.53 ± 4.3	4.86 ± 4.98	4.73 ± 4.95	2.33 ± 3.54	3.42 ± 4.07	3.48 ± 3.85	3.41 ± 3.95	3.36 ± 4.11	4.38 ± 4.49	2.84 ± 3.59	2.89 ± 3.57	4.49 ± 4.45
OREB	1.39 ± 1.8	2.75 ± 2.9	3.75 ± 3.43	1.1 ± 1.8	1.92 ± 2.46	3.24 ± 3.08	1.39 ± 1.97	2.22 ± 2.63	3.69 ± 3.69	1.16 ± 1.6	1.95 ± 2.23	3.32 ± 2.84
DREB	3.18 ± 2.87	5.38 ± 3.88	6.86 ± 4.73	3.4 ± 2.95	4.54 ± 3.78	6.06 ± 4.18	3.93 ± 3.18	5.14 ± 3.71	7.42 ± 4.51	3.6 ± 2.83	4.82 ± 3.54	6.43 ± 3.76
TREB	4.57 ± 3.36	8.14 ± 5.18	10.6 ± 6.21	4.5 ± 3.7	6.46 ± 5.14	9.29 ± 5.67	5.32 ± 3.88	7.36 ± 4.92	11.12 ± 6.2	4.82 ± 3.82	6.74 ± 4.35	9.73 ± 4.86
AST	3.51 ± 3.45	2.47 ± 2.62	1.75 ± 2.09	3.75 ± 3.99	2.49 ± 3.57	2.2 ± 2.59	4.22 ± 3.54	2.42 ± 2.63	1.92 ± 2.34	4.15 ± 3.34	2.17 ± 2.25	1.79 ± 2.2
PF	3.63 ± 3.05	3.92 ± 2.95	4.89 ± 3.41	2.94 ± 2.76	3.26 ± 2.82	3.88 ± 3.26	3.37 ± 2.91	3.8 ± 2.94	4.39 ± 3.33	3.87 ± 2.83	4.14 ± 2.85	4.73 ± 3.25
TOV	4.34 ± 3.34	3.72 ± 2.92	3.75 ± 3.13	3.11 ± 2.89	3.07 ± 2.82	3.48 ± 2.83	3.95 ± 3.09	2.87 ± 2.58	3.11 ± 2.72	3.08 ± 2.49	2.58 ± 2.22	3.09 ± 2.47
STL	2.2 ± 2.41	2.04 ± 2.28	1.56 ± 2.02	2.03 ± 2.24	1.81 ± 2.36	1.61 ± 2.05	2.05 ± 2.24	1.71 ± 2.02	1.28 ± 1.78	1.65 ± 1.87	1.3 ± 1.65	1.25 ± 1.78
BLK	0.17 ± 0.62	0.43 ± 1	1.06 ± 1.9	0.21 ± 0.69	0.5 ± 1.11	0.98 ± 1.6	0.22 ± 0.74	0.54 ± 1.14	1.08 ± 1.62	0.16 ± 0.56	0.5 ± 1.03	1.07 ± 1.64

Note: 2 ptM = Two-point made; 2 ptA = Two-point attempted; 3 ptM = Three-point made; 3 ptA = Three-point attempted; FTM = Free throws made; FTA = Free throws attempted; OREB = Offensive rebounds; DREB = Defensive rebounds; TREB = Total rebounds; AST = Assists; PF = Personal fouls; TOV = Turnovers; STL = Steals; BLK = Blocks.

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
