# Peer review of "The Regional Differences in Game-Play Styles Considering Playing Position in the FIBA Female Continental Basketball Competitions"

_ijerph, 2020, doi:10.3390/ijerph17165827_

Round 1
Reviewer 1 Report
The present work exposes a very interesting topic such as the comparison between players from different continents. In addition, it adds a new study to the analysis and knowledge of high-level women's basketball, which at the moment is scarce in the scientific literature. However, the analysis remains a little scarce with just the analysis of the situational variables of the game. These statistical data are closely related to each game model (contextual factors and team dynamics) or physical demands of each team. Therefore, it seems to have a very generic practical application without any key point.
Specific Comments
Line 29. Please change the size of citation numbers and brackets.
Line 30. Space is missing “players[6,7]”
Line 38. Consider to cite the sentence
Line 41. Space is missing “[10]applied”
Line 77. Space is missing “sample[3]”
Line 77. Please consider to change “our study” to an informal way of writing
Line 83. In my opinion validity and reliability section is not required here as you are just taking numbers already validated and published on web.
Line 10. Results. Consider to make bigger figures as it is difficult to read the results on them.
Line 166. Space is missing “percentage[3,23,23]”
Line 191. Space is missing “throws[26]”
Line 197. Space is missing “[23]suggested”
Line 210. You can delete the full name of KPI as it has been used previously in the text
Line 218. Space is missing “consensus[7]”
Line 223. Please consider to add as limitation the lack of physical demands variables to know a better description of differences among female player from different continents.
Line 231. Conclusion.
The conclusion is clear but should be taken with caution as none of the differences found among continents was bigger than “moderate”. This information opens a window for future research with more complex analysis to determine differences between different profiles on female basketball.
Author Response
Response to Reviewer 1 Comments
Point 1: Line 29. Please change the size of citation numbers and brackets.
Response 1: Revised as requested.
Point 2: Line 30. Space is missing “players[6,7]”
Response 2: Revised.
Point 3: Line 38. Consider to cite the sentence
Response 3: Based on your suggestion, I have added two references to cite the sentence.
- Sampaio, J.; Janeira, M.; Ibáñez, S.; Lorenzo, A. Discriminant analysis of game-related statistics between basketball guards, forwards and centres in three professional leagues. European journal of sport science 2006, 6, 173-178., J.; Janeira, M.; Ibáñez, S.; Lorenzo, A. Discriminant analysis of game-related statistics between basketball guards, forwards and centres in three professional leagues. European journal of sport science 2006, 6, 173-178.
- Sampaio, J.; Ibanez, S.J.; Gomez, M.A.; Lorenzo, A.; Ortega, E. Game location influences basketball players' performance across playing positions. International Journal of Sport Psychology 2008, 39, 205.
Point 4: Line 41. Space is missing “[10]applied”
Response 4: Revised.
Point 5: Line 77. Space is missing “sample[3]”
Response 5: Revised.
Point 6: Line 77. Please consider to change “our study” to an informal way of writing
Response 6: Based on your suggestion, I have changed the expression from “our study” to “in the present”.
Point 7: Line 83. In my opinion validity and reliability section is not required here as you are just taking numbers already validated and published on web.
Response 7: Thank you for your advising. The data is from the FIBA website but considering the accuracy and the standardization of academic research, I would like remain this part.
Point 8: Line 10. Results. Consider to make bigger figures as it is difficult to read the results on them.
Response 8: We do agree with your opinion. We now use the figure with “TIFF” format in the word, and it is convenient for reviewers to review this paper. We also upload the figure with “EPS” format in the system. It’s clear and bigger one.
Point 9: Line 166. Space is missing “percentage[3,23,23]”
Response 9: Revised.
Point 10: Line 191. Space is missing “throws[26]”
Response 10: Revised.
Point 11: Line 197. Space is missing “[23]suggested”
Response 11: Revised.
Point 12: Line 210. You can delete the full name of KPI as it has been used previously in the text
Response 12: Based on your suggestion. I have deleted the full name of KPI.
Point 13: Line 218. Space is missing “consensus [7]”
Response 13: Revised.
Point 14: Line 223. Please consider to add as limitation the lack of physical demands variables to know a better description of differences among female player from different continents.
Response 14: Thanks for your advice. I have added the “third limitation” considering the physical or physiology facts really have an important influence on player`s performance, which have not been mentioned in this study.
Point 15: Line 231. Conclusion.
The conclusion is clear but should be taken with caution as none of the differences found among continents was bigger than “moderate”. This information opens a window for future research with more complex analysis to determine differences between different profiles on female basketball.
Response 15: Thanks for your reminder. I have added the statement “although most of the differences were small or moderate, some game-play styles could still be identified” in the summary, which could give a simple tip for the readers.

Reviewer 2 Report
The authors wrote an interesting manuscript; it deals with continental differences between female playing positions. The theoretical background is relevant to the topic and provides general information. In line no.21 is incorrect citation (upper index). In lines no.30 and 41 are missing spaces between words and citations in brackets. In lines no.51, 52 and 58 is an incorrect citation. It should be “Štrumbelj et al.” and “Sampaio et al.” In line no.60, after the word “leagues” the full stop is missing.
Part “Material and Methods” is short but understandable. In line no.77 is missing space between “sample” and citation.
“Results” are presented in a well-arranged way. “Discussion” is apt, and “Conclusion” is comprehensible. However, there are some formal shortcomings (missing space between word and citations – lines no.166, 191, 197, and 218) and citations inconsistencies (should be: line 198 – Ibáñez et al.; line 207 – Paulauskas et al.; line 213 – Sampaio et al.).
Authors should explicitly state what could be an advantage (from a tactical point of view) when the teams from different continents would play against each other on tournaments like Basketball World Cup or Olympic games.
I appreciate part “Limitations,” where the authors are aware of the limitations of the study and the direction of future research in this area. As the authors stated, for a better picture of tactical differences between teams from different continents, the analysis of advanced statistics is needed.
Author Response
Response to Reviewer 2 Comments
Point 1: In line no.21 is incorrect citation (upper index).
Response 1: Thanks for your correction. Maybe the wrong citation is in line 29 and I have revised it.
Point 2: In lines no.30 and 41 are missing spaces between words and citations in brackets.
Response 2: Thanks, revised.
Point 3: In lines no.51, 52 and 58 is an incorrect citation. It should be “Štrumbelj et al.” and “Sampaio et al.”
Response 3: Thanks and revised.
Point 4: In line no.60, after the word “leagues” the full stop is missing.
Response 4: Revised.
Point 5: In line no.77 is missing space between “sample” and citation.
Response 5: Revised.
Point 6: In lines 166, 191, 197, and 218 are missing space between word and citations.
Response 6 : Revised.
Point 7: Citations inconsistencies (should be: line 198 – Ibáñez et al.; line 207 – Paulauskas et al.; line 213 – Sampaio et al.).
Response 7: Revised.
Point 8: Authors should explicitly state what could be an advantage (from a tactical point of view) when the teams from different continents would play against each other on tournaments like Basketball World Cup or Olympic games.
Response 8: Thanks for your suggestions. I have added some advice for coaches or players who play tournaments in line 188, line 211, line 226, line 236, and line 249.
Reviewer 3 Report
The article contains an interesting game analysis divided into playing positions. It should be emphasized that women's basketball is completely different game from men's basketball. Therefore, it requires separate analyzes.
Below I present a few detailed remarks necessary to be completed.
Major comments:
- in the introduction, please mention also the biomechanical analysis of the movement technique that is carried out on basketball players, eg. Struzik et al. 2014,
- complete the information with the name of representation of individual countries that were analyzed, divided into years,
- the discussion does not compare the obtained results with the works of other authors,
- supplement the discussion with a limitation section;
Minor comments:
- line 42: "body mass" instead of "weight".
Author Response
Response to Reviewer 3 Comments
Point 1: In the introduction, please mention also the biomechanical analysis of the movement technique that is carried out on basketball players, eg. Struzik et al. 2014,
Response 1: Thanks for your recommendation. Biomechanical analysis in basketball is really important, so I cited three articles (Struzik, et al; Kornecki et al; Pehar, et al) that mentioned the differences in shooting style and jump capacity of basketball players considering playing positions.
Point 2: Complete the information with the name of representation of individual countries that were analyzed, divided into years
Response 2: Thanks for your advice, but according to the author for instructions in this journal, the detailed information of individuals, countries, companies etc. need to be hided to protect their rights and interests.
Point 3: The discussion does not compare the obtained results with the works of other authors.
Response 3: Thanks for your suggestion. In fact, there are few studies on the technical performance of basketball players considering different positions in different regions. Only Sindik and Jukić (2011) and Sampaio, et al (2006). Therefore, it is hard to make a “point to point” comparison with previous results. In our study, we mentioned some research that could support our study in some degree.
In line 191, our results were linked with the prior studies from Erčulj and Štrumbelj, Paulauskas, et al. and Zhang, et al. pointed out that American player had better stature and fitness with high efficiency in terms of defensive rebounds because our results showed that American centers secured more defensive rebounds than European centers.
In line 213, we mentioned that our study was consistent with previously reported findings that African forwards tend to utilize free throws to scoring due to more energetic and active behavior in isolation (lay-ups & paint penetrations) which could lead to more physical contacts to free throws (Paulauskas, et al).
In line 220 we mentioned that European guards have a cautious attempt of 3-point shots which was partly supported by Ibáñez et al and Madarame.
In line 234, we mentioned that guards play a key role in high-intensity competitions (e.g. Europe & America) and perform a higher number of AST than that of Africa, which was partly supported by Madarame, Paulauskas, et al. and García, et al. who suggested that offense in Europe was well-organized.
In line 239, we also mentioned that African players from all of positions made more TOV compared with the corresponding positional players from other continental championships which were linked to the studies of Sampaio, et al. and Madarame suggested that the game pace and the unorganized manner of play in Africa were the main reasons that lead to the emergence of turnovers.
Point 4: Supplement the discussion with a limitation section
Response 4: “Limitation” has already inserted in this article, which locate just above the “conclusion”.
Point 5: Line 42: "body mass" instead of "weight".
Response 5: Thanks and revised.

Round 2
Reviewer 3 Report
The manuscript was revised my corrections request in the first round.